

# Technical Note: Improving the computational efficiency of sparse matrix multiplication in linear atmospheric inverse problems

Vineet Yadav[1], Anna M. Michalak[2]

[1]Jet Propulsion Laboratory, California Institute of Technology, Pasadena, California, 91011, USA
[2]Department of Global Ecology, Carnegie Institution for Science, Stanford, California, 94305, USA

*Correspondence to*: Vineet Yadav (vineet.yadav@jpl.nasa.gov)

**Abstract.** Matrix multiplication of two sparse matrices is a fundamental operation in linear Bayesian inverse problems for computing covariance matrices of observations and *a posteriori* uncertainties. Applications of sparse-sparse matrix multiplication algorithms for specific use-cases in such inverse problems remain unexplored. Here we present a hybrid-
parallel sparse-sparse matrix multiplication approach that is more efficient by a third in terms of execution time and operation count relative to standard sparse matrix multiplication algorithms available in most libraries. Two modifications of this hybrid-parallel algorithm are also proposed for the types of operations typical of atmospheric inverse problems, which further reduce the cost of sparse matrix multiplication by yielding only upper triangular and/or dense matrices.

## 1 Introduction

Sparse-Sparse (SS) matrix multiplication forms the computational backbone of scientific computation in many fields. In this study, applications of SS matrix multiplication to the solution of linear Bayesian inverse problems aimed at estimating greenhouse gas emissions are presented (for details on the methodological formulation of these inverse problems, see Enting 2005). However, the issues associated with SS matrix multiplication discussed here have wide implications for many fields of research.

Most of the SS matrix multiplication algorithms that can be used with any sparse matrix structure (e.g., diagonal, block-diagonal, Toeplitz, Binary and Vandermonde matrices) use modifications of the algorithm proposed by Gustavson (1978). Parallel implementation of this algorithm for matrices stored in compressed sparse row (CSR) format for CPUs and GPUs appear in standard libraries (NVIDIA 2016; Intel 2016a). For CPUs, these general-purpose parallel implementations either (1) precompute the number of non-zero entries in the output matrix (e.g. a matrix $\mathbf{AB}_{(r \times t)}$ that results from the multiplication

of a matrix $\mathbf{A}_{(r \times t)}$ by a matrix $\mathbf{B}_{(t \times t)}$) and then compute matrix-matrix product (for details see, Math Kernel Library routine "mkl_csrmultcsr", Intel 2016b), or (2) progressively adjust the space required to store the output matrix (for details see; Davis 2006; Gilbert, Moler, and Schreiber 1992). Alternatively, on a GPU, a sufficiently large non-adjustable memory (a.k.a. Random Access Memory) is assigned to store the output of SS matrix multiplication (variations of this approach are proposed in Liu and Vinter 2014). Other approaches for increasing the computational efficiency of SS matrix multiplication

have focused on specific applications, such as SS matrix multiplication for hyper-sparse matrices (Buluç and Gilbert 2010),





multigrid PDEs (Mccourt et al., 2013), or multiplication of a binary matrix with a dense matrix (Blelloch et al., 2008), among others.

The development of SS matrix multiplication approaches tailored to improving the computational efficiency of atmospheric inverse problems has not been previously explored. Here, we address this need and propose (1) a single pass hybrid-parallel algorithm for performing matrix-matrix multiplication on CPUs and describe conditions under which this approach can be computationally more efficient than a standard dual pass algorithm, (2) a modification of the Gustavson's (1978) algorithm for obtaining the upper or lower triangular portion of a symmetric sparse matrix, and (3) a modification of the Gustavson's (1978) algorithm for obtaining a dense matrix from the multiplication of two sparse matrices. In lieu of providing pseudo code, the source code for all the proposed algorithms is provided in supplementary material. This source code can be compiled and directly incorporated into linear atmospheric inverse modeling procedures.

## 2 Application of sparse-sparse matrix multiplication in inverse problems

In a previous work, Yadav and Michalak (2013) presented an efficient method for multiplying two matrices in the context of inverse problems when one of the matrices could be expressed as a Kronecker product of two smaller matrices. This form of matrix multiplication occurs in inverse problems when the Jacobian of the forward problem (i.e. the sensitivity matrix of the elements of the observation vector to the elements of the state vector) $\mathbf{H}_{(n \times m)}$ (for details regarding $\mathbf{H}$ , see Michalak, Bruhwiler and Tans 2004) is multiplied by a separable covariance matrix $\mathbf{Q}_{(m \times m)}$ that describes the decorrelation of prior uncertainties in space and in time. In this special case, the Yadav and Michalak (2013) algorithm accommodates all possible combinations of sparse and dense matrix multiplication (e.g. sparse-sparse, dense-dense, dense-sparse and sparse-dense). However, this method does not improve the computational efficiency of matrix multiplication (a) when $\mathbf{Q}$ is sparse and cannot be expressed as a Kronecker product, or (b) when a matrix $\mathbf{HQ}$ that does not have any special structure needs to be multiplied by $\mathbf{H}^T$ , which is another common operation in inverse problems. In the first case, the matrix multiplication can be performed through a general sparse matrix routine, whereas in the second case computational efficiency can be improved by considering the properties of $\mathbf{HQH}^T$ , such as the fact that matrix multiplication of $\mathbf{H}$ with a symmetric matrix $\mathbf{Q}$ followed by multiplication by $\mathbf{H}^T$ results in a symmetric matrix ( $\mathbf{HQH}^T$ ). Thus, computational efficiency can be gained by computing only the upper or lower triangular portions of $\mathbf{HQH}^T$ . Additional, computational efficiency can be gained when it is known that the output of the matrix multiplication of two SS matrices will yield a dense matrix. Such a case is common in linear inverse problems because the multiplication of $\mathbf{H}$ by $\mathbf{Q}$ reduces sparsity, and therefore $\mathbf{HQ}$ , when multiplied by $\mathbf{H}^T$ results in a dense symmetric matrix. In these situations, pre-allocation of memory for $\mathbf{HQH}^T$ is straightforward, and if only the upper or lower triangular matrix is required, then obtaining a dense matrix from the multiplication of two sparse matrices is even more efficient. Routines for obtaining a triangular matrix as an output from the matrix multiplication of two dense





matrices already appear in the Math Kernel Library and therefore development of an algorithm to accommodate this situation for inverse problems is not required (for details see; A Matrix Multiplication Routine That Updates Only the Upper or Lower Triangular Part of the Result Matrix, Intel (2016c)).

Thus, algorithmically, SS multiplication in inverse problems manifests itself in three possible forms:

(a) Multiplication of a sparse matrix $\mathbf{H}$ with a sparse diagonal, sparse block-diagonal, or full-dense covariance matrix $\mathbf{Q}$

      (b) Multiplication of a sparse $\mathbf{HQ}$ with a sparse $\mathbf{H}^T$ that result in a sparse symmetric matrix

      (c) Multiplication of a sparse $\mathbf{HQ}$ with a sparse $\mathbf{H}^T$ that result in a dense symmetric matrix

Each of these three forms of matrix multiplication requires a distinct algorithmic treatment for increasing computational

efficiency. The names of the routines that perform these three forms of SS matrix multiplication, as given in supplementary material, are shown in Figure 1 in terms of matrices $\mathbf{HQ}$ and $\mathbf{H}^T$. Note that even though the three forms of matrix multiplication described above are given in terms of atmospheric inverse problems, they are also common in other applications (e.g., Cressie and Johannesson, 2008).

## 3 Classical approach for multiplication of two sparse matrices

The classical approach for matrix multiplication of two SS matrices involves either a single pass or a double pass algorithm. This approach relies on keeping track of indices of non-zero entries in the two matrices to be multiplied. The operations involved are shown in Appendix 1 in the function "dense_nosym." Note that this function does not depict automatic adjustment of the memory and results in a dense matrix. In the single pass version of the classical algorithm (e.g. adopted in Matlab (Gilbert, Moler, and Schreiber (1992)), a set amount of memory is assigned for storing the output of the matrix

multiplication, which is then adjusted if the size of the output exceeds the assigned memory. On the other hand, in a double pass algorithm, the number of non-zero entries in the output matrix is first computed (this is also referred to as symbolic matrix multiplication), after which SS matrix multiplication is performed.

### 3.1 Parallel approach for multiplying two sparse matrices

For the purpose of application to atmospheric inverse problems, we parallelize the classical single pass algorithm and divide

work of multiplying two SS matrices across threads and nodes by rows. This can be defined as 1-dimensional (1D) matrix multiplication, as partitioning of the matrix multiplication between two conforming matrices happens on the basis of rows of the first matrix (see Figure 2). It is similar to 1D row-wise SS matrix multiplication as discussed in, Ballard et al. (2016). However, Ballard et al. (2016) do not provide a methodology for computing non-zero entries in the output matrix, which in our approach can be specifically tailored for inverse problems.

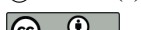


Even though the 1D algorithm implemented here can be used for general SS matrix multiplication, it is most suited for multiplying $\mathbf{H}$ and $\mathbf{Q}$ or $\mathbf{HQ}$ and $\mathbf{H}^T$ in linear Bayesian inverse problems, as: (1) each row in $\mathbf{H}$ in these problems is guaranteed to have non-zero entries, which allows good load balancing across nodes and threads, and (2) the memory for storing non zero entries in the output matrix can be computed efficiently if the structure of the covariance matrix $\mathbf{Q}$ is pre-

specified (e.g., diagonal, block-diagonal or full) as allowed in the routines given in supplementary material.

The application of the 1D approach for SS matrix multiplication in a hybrid environment is exemplified in Figure 2, where an arbitrary ($r \times t$) matrix $\mathbf{A}$ is multiplied by a ($t \times t$) matrix $\mathbf{B}$ using two nodes. In this case, each of the two nodes gets $r/2$ rows of $\mathbf{A}$ to multiply with matrix $\mathbf{B}$. This task is further divided across two threads on each node and in the final step results are collected to obtain the ($r \times t$) matrix $\mathbf{AB}$. As parallel SS matrix multiplication is communication-limited,

performing this computation on a single node by using multiple threads is much faster relative to SS matrix multiplication accomplished by using a single thread across multiple nodes. This happens because matrices $\mathbf{A}$ and $\mathbf{B}$, as shown in Figure 2, are shared by threads and the only private variables are the partitioned output of the matrix multiplication (e.g. $\mathbf{A_1 B}$, $\mathbf{A_2 B}$, $\mathbf{A_3 B}$ and $\mathbf{A_4 B}$).

The ID approach is simple; however, knowing the number of non-zero entries in the output matrix can significantly reduce

the computational cost of SS matrix multiplication, as it allows efficient pre-allocation of memory for storing the output of matrix multiplication. In linear Bayesian inverse problems, even though the exact number of non-zero entries in the output matrix is unknown, an estimate can be obtained by looking at the structure of $\mathbf{Q}$ and the sensitivity matrix $\mathbf{H}$, which is sparse and banded in the case of regional inversions and where the elements of the state space only influence observations over a finite period of time (see Figure 3; for details on regional inversions see Gourdji et. al., 2010). For example, when $\mathbf{Q}$

is a diagonal matrix, the number of non-zero entries in $\mathbf{HQ}$ would be equal to non-zero entries in $\mathbf{H}$. Due to the sparsity structure of $\mathbf{H}$ in regional atmospheric inverse problems, approximate estimates of non-zero entries in $\mathbf{HQ}$ can also be obtained for dense and block-diagonal $\mathbf{Q}$ as shown in Table 1. Since $\mathbf{HQH}^T$ is symmetric and most likely to be dense or have a significant proportion of non-zero entries, it should always be obtained as dense upper or lower triangular matrix. For inversions where elements of the state space indefinitely impact later observations (e.g. global atmospheric inversions), $\mathbf{H}$ is

denser than in the regional inversion case, and a default formulation as discussed in the next paragraph should be used for multiplying $\mathbf{H}$ and $\mathbf{Q}$.

In supplementary material, we show the application of 1D SS matrix multiplication algorithm in a multithreaded framework, which can be easily extended to a hybrid computing environment consisting of multiple nodes and multiple threads. If the covariance structure of $\mathbf{Q}$ is specified, then we use Table 1 to compute non-zero entries in the output matrix. In case it is not

specified, the algorithm pre-allocates the number of non-zero entries in the output matrix as a percentage. In the default case, it is assumed that the output matrix would have 10% non-zero entries (for e.g. $0.10rt$ for storing $\mathbf{AB}_{(r \times t)}$ that is obtained by



multiplying $\mathbf{A}_{(r \times t)}$ and $\mathbf{B}_{(t \times t)}$ ) divided across nodes and/or threads. This percentage can be overridden and lowered. As implemented in the source code, if it is instead specified as > 10% then an upper limit of 10% is set and adjusted after this threshold is crossed i.e., a larger amount of memory is required. In automated mode, the size of the memory is doubled after the threshold is reached; however, this can also be overridden. If a lower amount of memory is utilized then what is

specified, then we adjust the memory at the end of the matrix multiplication.

## 3.2 Performance of the algorithm

The performance of the 1D SS matrix multiplication algorithm is evaluated by comparing it with the "mkl_csrmultcsr" routine provided in Intel's MKL library. A direct comparison of the computational efficiency of the proposed algorithms with the algorithms designed for GPUs is not performed as GPUs are not universally available, considerable time, for

performing multiplication of two SS matrices is used for transferring data on a GPU, and most of the large inverse problems require out of core implementation that makes SS matrix multiplication inefficient on GPUs.

A real application from an atmospheric inverse problem is used to assess the performance of the two algorithms. In this application, double-precision matrices, $\mathbf{HQ}$ and $\mathbf{H}^T$ of dimensions ($n \times m$) and ($m \times n$) where $n = 1070$, $m = 10^7$, $NNZ_{\mathbf{HQ}} =$

$NNZ_{\mathbf{H}^T} = 8.55 \times 10^7$ are multiplied in a multithreaded environment to yield a dense ($n \times n$) symmetric matrix with $1070^2$

$= 1.14 \times 10^6$ non-zero entries. Even though computing an upper or lower triangular matrix would suffice for this problem, a full dense symmetric matrix was obtained in CSR form and the number of non-zero entries in the output matrix was not specified. The performance of the two algorithms was assessed on an Intel Xeon E5440 Harpertown 2.83 Ghz computer with 12MB L2 Cache. They were compared on three metrics, namely the: (1) floating-point operations (FLOPS) required, (2) time taken for execution, and, (3) peak and total memory usage. The results presented in Table 2 show that the proposed

algorithm is approximately 33% faster in execution time and takes approximately 38% fewer FLOPS than the "mkl_csrmultcsr" routine included in Intel's MKL. However, as the 1D parallel algorithm does not know the exact number of non-zeros in the output matrix, the total and peak memory usage is higher than for "mkl_csrmultcsr", which requires an exact count of the number of non-zero entries in the output matrix. The difference in the performance of the 1D parallel algorithm and "mkl_csrmultcsr" is primarily due to the elimination of the time required to compute non-zero entries in the

output matrix. These differences become even more pronounced when only the upper or lower triangular sparse or dense matrix is required from the matrix multiplication of sparse $\mathbf{HQ}$ and $\mathbf{H}^T$. Evaluation of the computational efficiency for obtaining these triangular matrices was not performed as the FLOPS required for computing the matrix product in these cases is substantially lower than "mkl_csrmultcsr". Moreover, if the dense triangular product is obtained then the need to compute number of non-zero entries is also eliminated.



## 4 Conclusion

Matrix multiplication remains a critical bottleneck in the solution of large-scale linear inverse problems and for many other scientific applications, and improving the computational efficiency of matrix multiplication is therefore an active area of research. A practical algorithm that can reduce the asymptotic bounds of matrix multiplication below $O(n^{2.81})$ remains

elusive. However, novel algorithms are being proposed for eliminating this bottleneck for specific classes of matrices. The methods for SS matrix multiplication proposed here increase the computational efficiency of atmospheric inversions by accounting for the sparsity structure of the sensitivity matrix and *a priori* covariance matrix. Three different formulations of the SS matrix multiplication are proposed. The source code for each of these is provided for direct integration in software designed for the solution of inverse problems. Hardware-specific optimizations can further improve the performance of the

proposed algorithms. Updates to the algorithms proposed in this work would be required for porting them on new computer architectures and would be explored as part of further research.

### Source Code Availability

All source code required to implement the methods proposed in this research is available in supplementary material. The

dependency graph for the functions included in the source code is shown in Appendix 2.

### Acknowledgement

This work was supported by funds from National Science Foundation under Grant No. 1342076. Most of the research was carried out at the Jet Propulsion Laboratory, California Institute of Technology, under a contract NNN15R040T between

Carnegie Institution of Washington and National Aeronautics and Space Administration.



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





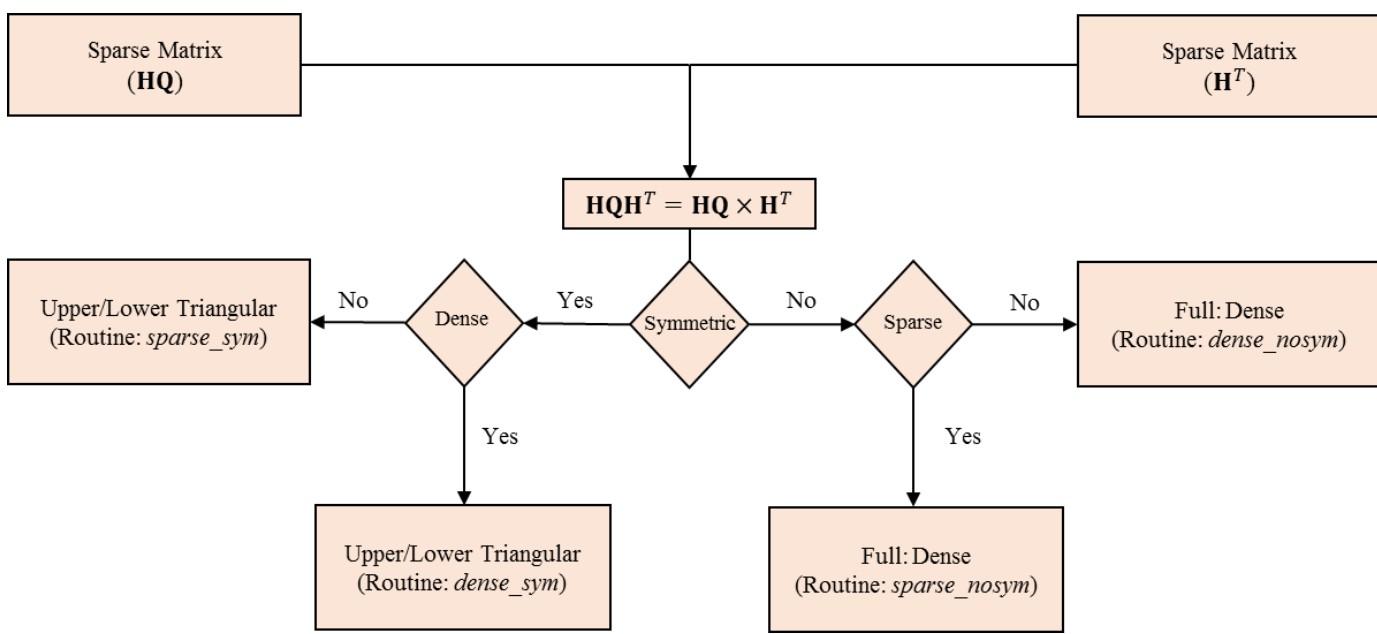

**Figure 1: Software routines as given in supplementary material for sparse-sparse matrix multiplication in inverse problems.**





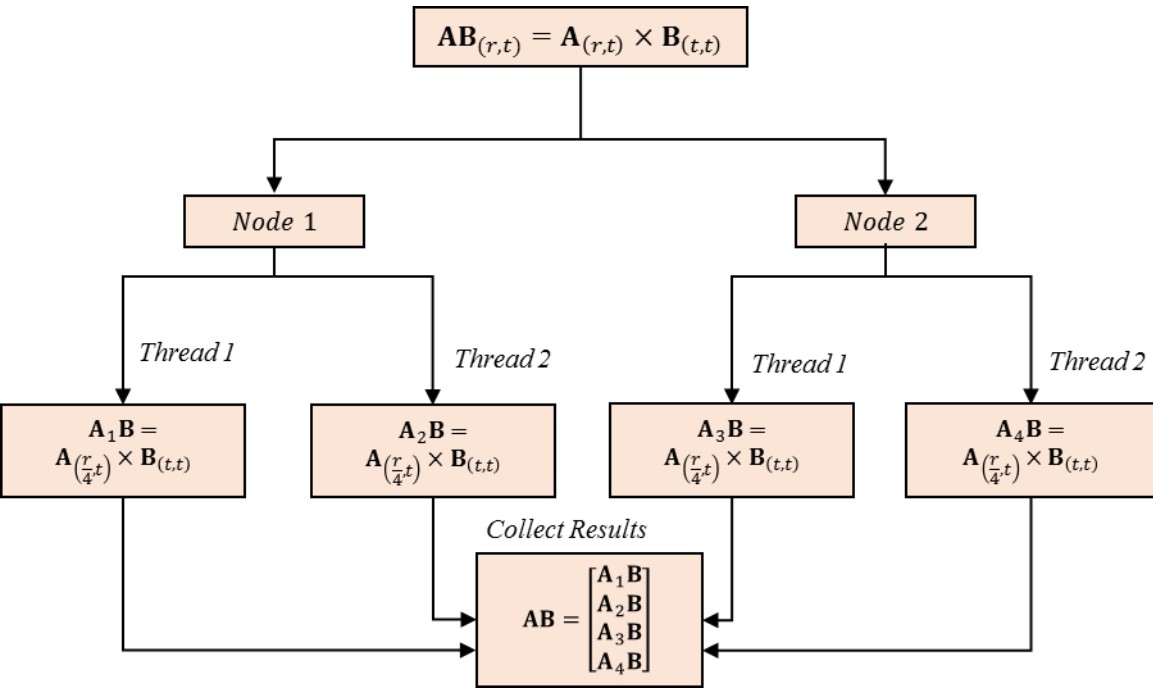

**Figure 2: Example of one-dimensional hybrid parallel sparse-sparse matrix multiplication. Note that in this figure it is assumed that $\mathbf{A}$ has even number rows. In case there are odd number of rows in $\mathbf{A}$ than the 1D parallel algorithm divides them across nodes and threads as evenly as possible.**



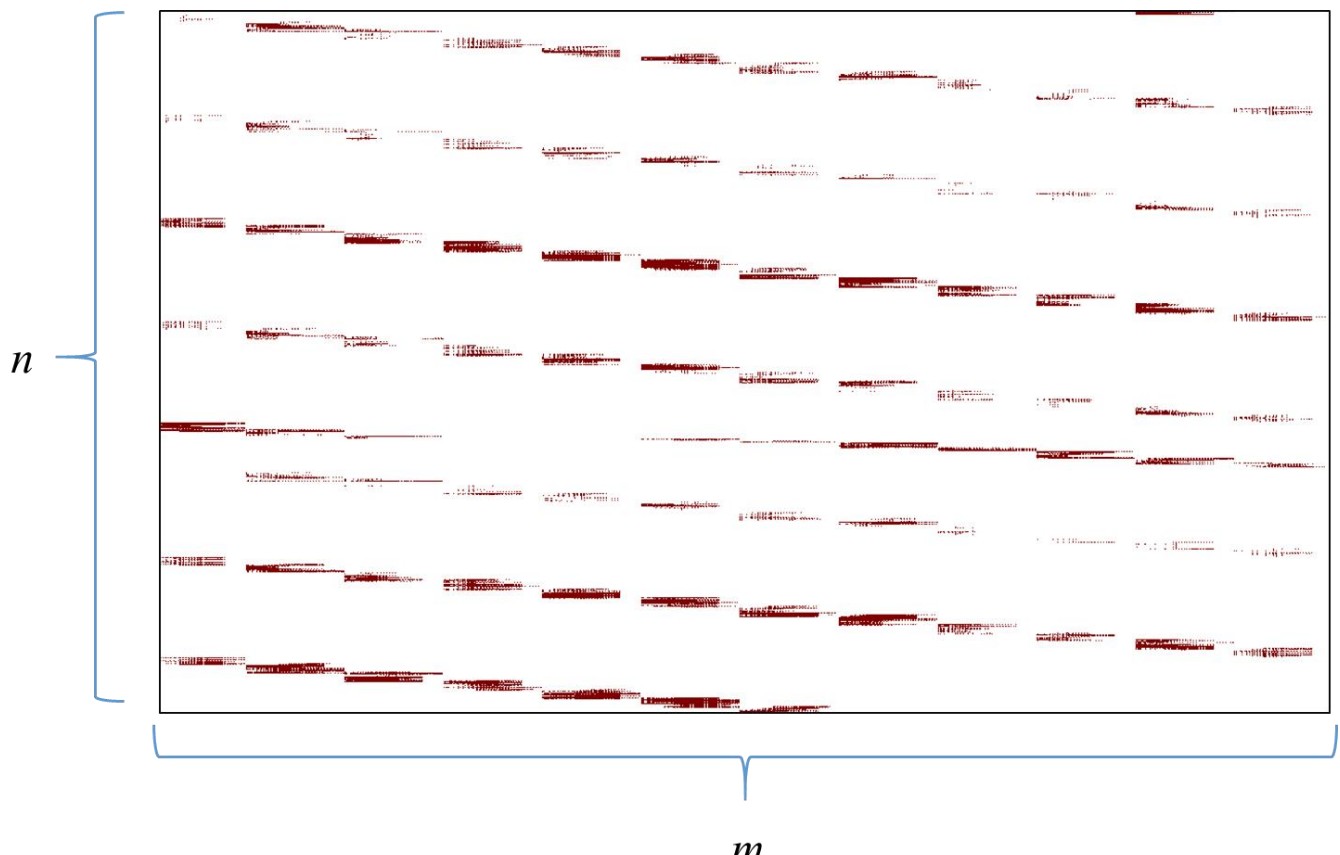

$n$

$m$

**Figure 3: Non-zero entries (brown) in a typical sensitivity matrix ( H ) used in regional atmospheric inversions.**



| Sparsity Structure of $\mathbf{H}$ | Sparsity Structure of $\mathbf{Q}$ | $NNZ_{\mathbf{HQ}}$ |
|---|---|---|
| Banded | Diagonal | $= m$ |
| Banded | Block-diagonal | $\approx (n \times m)/k$ |
| Banded | Full | $= n \times m$ |

**Table 1: Estimate of non-zero entries in $\mathbf{HQ}$ based on banded sparsity structure of $\mathbf{H}$ in regional inversions (see figure 3) and different formulations of covariance structure $\mathbf{Q}$. Note k in second row and third column stands for number of blocks in the block-diagonal matrix $\mathbf{Q}$.**

20

25



| Metrics | Performance of the proposed algorithm | Performance of MKL algorithm |
|---|---|---|
| FLOPS | $\approx 1.53 \times 10^{10}$ | $\approx 2.45 \times 10^{10}$ billion |
| Time Taken for Execution | $\approx 28$ seconds | $\approx 42$ seconds |
| Peak Memory Consumed | $\approx 2.45$ Gb | $\approx 2.57$ Gb |
| Total Memory Consumed | $\approx 4.88$ Gb | $\approx 3.19$ Gb |

**Table 2: Comparative performance of 1D sparse-sparse matrix multiplication algorithm with Math Kernel Library routine mkl_csrmultcsr.**

20





**Appendix 1**

Source code for obtaining dense matrices from multiplication of two sparse matrices. For clarity we provide a reference matrix **p** and its storage in compressed sparse row format. The source code given below also require that each input matrix specifies the number of rows and columns in the matrix and the number on non-zero entries in the input matrix

---

**Insert 1:** Example of a compressed sparse row (CSR) format for storing a sparse matrix. The variables "rowPtr", "colInd", "values", "nzmax", "rows" and "cols" are used to define additional properties of a sparse matrix stored in CSR form in the source code given below.

$$\mathbf{p} = \begin{bmatrix} 1 & -1 & 0 & -3 \\ -2 & 5 & 0 & 0 \\ 0 & 0 & 4 & 6 \\ -4 & 0 & 2 & 7 \end{bmatrix}$$

| rowPtr | = | 1 | 4 | 6 | 8 | 11 | | | | | |
|--------|---|---|---|---|---|----|---|---|---|---|---|
| colInd | = | 1 | 2 | 4 | 1 | 2 | 3 | 4 | 1 | 3 | 4 |
| values | = | 1 | -1 | -3 | -2 | 5 | 4 | 6 | -4 | 2 | 7 |

nzmax = 10 (Number of non-zeros in matrix **p** )

rows = 4 (Number of rows in matrix **p** )

cols = 4  (Number of rows in matrix **p** )

---

```
/* structure "sparsemat" is used to store a precision sparse matrix in CSR form. For details regarding CSR form
see Insert 1 given above*/
```

```
struct sparsemat {
       /*
       nzmax = number of non zeros in a sparse matrix stored in compressed sparse row (CSR) format
       rows  = number of rows in a sparse matrix stored in CSR form
       cols  = number of cols in a sparse matrix stored in CSR form
```
15
```
       rowPtr = number of non zero entries in each row of a sparse matrix
       in a cumulative form, where number of non-zeros in any row of the sparse matrix
       can be obtained by subtracting rowPtr[j+1]-rowPtr[j]. Here j represents the row for which
       we want to obtain non-zero entries. For details see Insert 1.
       colInd = column number for non-zero entries in a sparse matrix stored in CSR form
```
20
```
       values = value of non-zero entries in the sparse matrix stored in CSR form
       */
       int nzmax;
       int rows;
```





```
             int cols;
             int *rowPtr;
             int *colInd;
             double *values;
5       };

        /* structure "darray" is used to store a dense double matrix, its rows and columns are specified in variables
        rows and cols.*/

struct darray
        {
             double * array;
             int rows;
             int cols;
};

        void dense_nosym(const struct sparsemat * const matrixa, const struct sparsemat * const matrixb, struct darray *
        const matrixc)

/* This routine multiplies two sparse matrices stored in CSR form
        with zero based indexing (i.e. number of rows and number of columns
        are numbered from 0 to rows -1 and 0 to cols -1 respectively)and produces
        a dense upper triangular matrix based on the assumption that multiplication
        of two sparse matrices results in a symmetric matrix*/
{
             int i, j, k, col_num_a, col_num_b, konstant;
             double value;
             matrixc->rows = matrixa->rows;
             matrixc->cols = matrixb->cols;
30           matrixc->array = (double *)calloc(matrixa->rows*matrixb->cols, sizeof(double));
        # pragma omp parallel for private (i,j,k,value,col_num_a,col_num_b,konstant)
             for (i = 0; i < matrixa->rows; i++)
             {
                     konstant = i*matrixa->rows;
35                   for (j = matrixa->rowPtr[i]; j <= matrixa->rowPtr[i + 1] - 1; j++)
                     {
                             value = matrixa->values[j];
                             col_num_a = matrixa->colInd[j];
                             for (k = matrixb->rowPtr[col_num_a];
40                                  k <= matrixb->rowPtr[col_num_a + 1] - 1; k++)
                             {
                                     col_num_b = matrixb->colInd[k];
                                     matrixc->array[konstant + col_num_b] += value * matrixb->values[k];
                             }
45                   }

             }
        }

50
```





## Appendix 2

In supplementary material, source code written in C++ for four routines mentioned in figure 1 is given. A makefile is also given as part of the source code. This makefile can be used to compile the source code. Small matrices are included as part of the main function for testing the source code. Another main function is included in the source code directory which can be

5    used in place of the "main-test" function to read sparse matrices from the file and write the output of the multiplication to a file. The file based main function can be used to read large matrices. The dependency graph for the source code for the calls made from the main function for testing various formulations of SS matrix multiplication is shown below (for more details see the main function in the supplementary material)

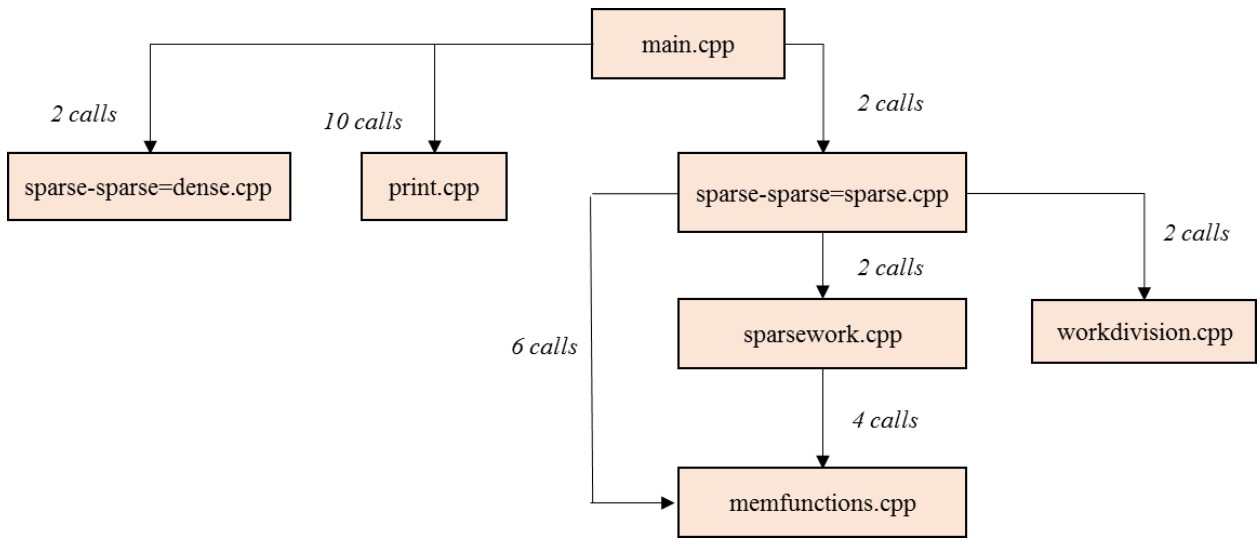

