# Peer review of "Technical Note: Improving the computational efficiency of sparse matrix multiplication in linear atmospheric inverse problems"

_Geoscientific Model Development, 2016_

## Referee Comment (RC1) · Anonymous Referee #1 · 21 Oct 2016

The authors present a way to accelerate sparse matrix-matrix multiplication which can
be a major bottleneck in the course of solving inverse problems, using a specific for-
mulation. The paper is short, well written and self-contained.

The paper only provides matrix–matrix multiplication. As it stands, I find the novelty of
this paper to be quite limited. There are other parallel matrix libraries that can perform
in similar computational environments (although they would might require MPI) such
as SCALAPACK, PETSc, FLAME, Elemental, etc. Why should one use this package
as opposed to these other packages?

I think the benefit of this package maybe the examples briefly mentioned in Section
2. This paper may be considered for publication if additional software developed is

performed to construct different types of covariance matrices (and their appropriate data structures) and handle different types of forward operators. For example, how can one pre-compute the sparsity pattern in inverse problems? More insight into these issues maybe more valuable.

General Comments: The numerical comparisons are limited in depth and breadth which weakens the contributions of this paper.

1. The algorithm proposed uses OpenMP but intel-mkl libraries are multi-threaded. Why is comparison between the two methods fair since they use different software paradigms? Please provide some justification for this.

2. Timing results are only provided for only one realization of $m, n$ and one particular sparsity pattern. Are similar gains to be found for different sparsity patterns, or problem sizes? Since this is claimed to be a general purpose algorithm for inverse problems and matrix–matrix multiplication, either there should be a detailed analysis of computational cost, or extensive numerical testing for different choices of $m, n$ and sparsity pattern.

Minor Comments:

1. In the introduction, matrix–matrix multiplication is mentioned as a fundamental operator in Bayesian inverse problems. This is only true for one particular computational formulation of the inverse problem; an alternative approach is to solve the MAP estimate using iterative methods, for which matrix–matrix multiplication may not be necessary nor desired. Please change the introduction to reflect this.

2. It would be useful if you could provide the inputs to the routines in Figure 1.

To conclude, this paper is not convincing either in its novelty or its execution. However, there may be some benefit to users if this package is more user friendly to application

specialists and careful numerical experiments are performed to illustrate the computational benefits over intel mkl and other packages.

---

## Referee Comment (RC2) · Anonymous Referee #2 · 24 Jan 2017

This technical note by the authors is an attempt to improve efficiency of a common operation in inverse problems which is Sparse-Sparse matrix multiplication. Although the paper is well structured, it lacks significantly on the experiments performed and presentation of the results that could prove the robustness and absolute efficiency of the proposed algorithm. The authors need to have the following experiments performed in order to be assertive about the efficiency of their algorithm.

1. It is suggested to produce a plot of number of cores versus speed up for intel mkl and proposed algorithm

2. There should be a table representing processing and memory performance for various sizes of matrices and sparsity patterns for both algorithms

**GMDD**

3. For each of these matrix types (size, sparsity), the authors need to furnish the number of threads spawned as it provides a deeper insight into the performance of the algorithms

4. The authors should also provide the parameters that were passed to the intel mkl csrmultcsr routine such as sort=3 reorders both input matrices and the output matrix which is time consuming

5. It is also suggested to compare the proposed algorithm against other available libraries/algorithms such as SPGEMM, FLAME, BLAS, LAPACK